# Subgenomic Flaviviral RNAs of Dengue Viruses

**DOI:** 10.3390/v15122306

**Published:** 2023-11-24

**Authors:** Yi Liu, Wuxiang Guan, Haibin Liu

**Affiliations:** 1Hubei Jiangxia Laboratory, Wuhan 430200, China; 2Center for Emerging Infectious Diseases, Wuhan Institute of Virology, Center for Biosafety Mega-Science, Chinese Academy of Sciences, Wuhan 430207, China

**Keywords:** sfRNA, replication, Xrn1-resistant RNA (xrRNA), 3′ UTR, Dengue virus

## Abstract

Subgenomic flaviviral RNAs (sfRNAs) are produced during flavivirus infections in both arthropod and vertebrate cells. They are undegraded products originating from the viral 3′ untranslated region (3′ UTR), a result of the action of the host 5′-3′ exoribonuclease, Xrn1, when it encounters specific RNA structures known as Xrn1-resistant RNAs (xrRNAs) within the viral 3′ UTR. Dengue viruses generate three to four distinct species of sfRNAs through the presence of two xrRNAs and two dumbbell structures (DBs). The tertiary structures of xrRNAs have been characterized to form a ringlike structure around the 5′ end of the viral RNA, effectively inhibiting the activity of Xrn1. The most important role of DENV sfRNAs is to inhibit host antiviral responses by interacting with viral and host proteins, thereby influencing viral pathogenicity, replicative fitness, epidemiological fitness, and transmission. In this review, we aimed to summarize the biogenesis, structures, and functions of DENV sfRNAs, exploring their implications for viral interference.

## 1. Introduction

Flaviviruses, a genus of enveloped, positive, single-stranded RNA viruses belonging to the family Flaviviridae, include important human pathogens like Dengue virus (DENV), Japanese encephalitis virus (JEV), West Nile virus (WNV), Zika virus (ZIKV), yellow fever virus (YFV), and tick-borne encephalitis virus (TBEV). These viruses cycle between arthropod vectors and vertebrate hosts, causing severe diseases and high mortality rates [1]. DENV, considered a critical flavivirus, was listed by WHO as one of the ten potential threats to global health in 2019. It is endemic in over 100 countries, especially in tropical and sub-tropical regions of South-East Asia, North America, and South America. Approximately two-fifths of the world population are at risk of DENV, resulting in about 390 million infections and 20,000 deaths annually [2]. However, effective antiviral treatments and vaccines are still lacking [3].

As DENV poses a serious life threat, extensive efforts have been made in the last 80 years to understand the virus’s life cycle and host interactions [4,5]. The DENV life cycle including entry, uncoating, translation, RNA synthesis, assembly, and release has been elucidated [6,7,8,9,10,11], with numerous steps being considered as potential antiviral targets [1].

Besides the full genome, DENV also produces a subgenomic RNA, known as subgenomic flaviviral RNA (sfRNA), during viral RNA replication [12,13]. This sfRNA is a 3′ terminal degradation product of viral RNA, generated by host 5′-3′ exoribonuclease. It is highly conserved in the family Flaviviridae [12,13,14]. Although sfRNA-deficient flaviviruses can be rescued in vitro and in vivo, sfRNA is critical for the efficient viral replication of all known pathogenic flaviviruses [15].

**Figure 1 viruses-15-02306-f001:**
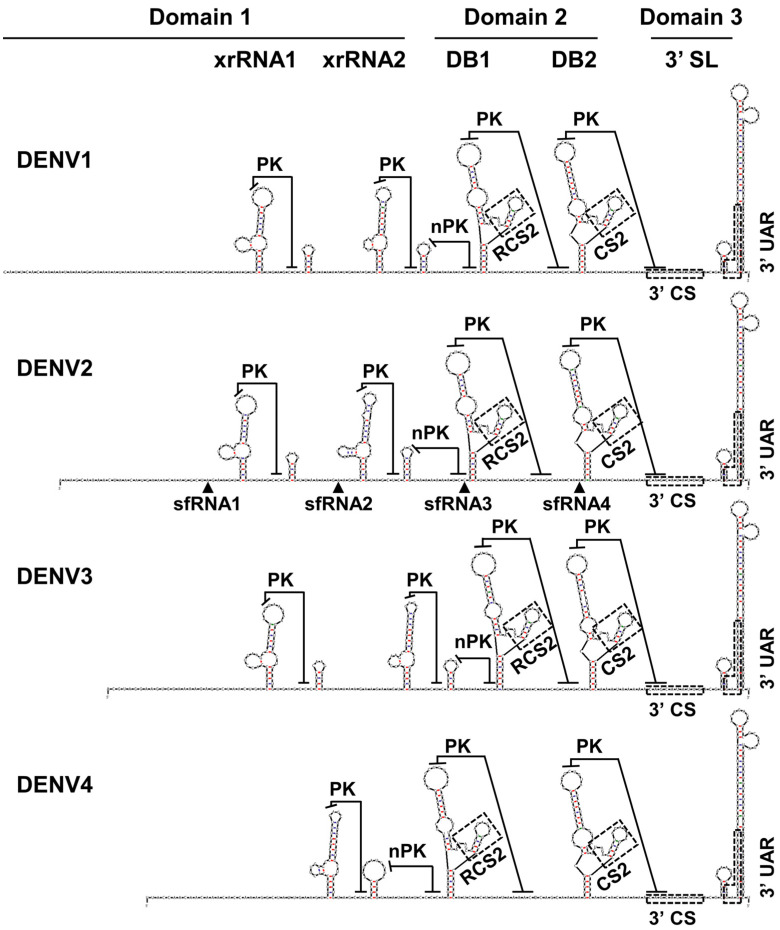
The structures of the 3′ untranslated region (3′ UTR) of Dengue viruses (DENV). Diagrams of 3′ UTRs are assembled by individual secondary structures predicted and folded by mFold (http://www.mfold.org/mfold/applications/rna-folding-form.php, accessed on 21 September 2023). The 3′ UTR sequences of DENV1-4 are from NCBI RefSeq assembly (DENV1 3′ UTR, 462 bp, GCF_000862125.1; DENV2 3′ UTR, 451 bp, GCF_000871845.1; DENV3 3′ UTR, 440 bp, GCF_000866625.1; DENV4 3′ UTR, 384 bp, GCF_000865065.1). Xrn1-resistant RNA(XrRNA), pseudoknot (PK), novel PK (nPK), conserved sequence 2 (CS2), repeat conserved sequence 2 (RCS2), and two conserved sequences for genome cyclization, 3′ conserved sequences (CS) and 3′ upstream AUG region (UAR), are indicated. The halt sites of sfRNA1, sfRNA2, sfRNA3, and sfRNA4 are indicated by arrows [16,17].

sfRNA is classified as a noncoding RNA, originating from the 3′ untranslated region (3′ UTR). Thus, its primary function is to interact with numerous host factors, especially those involved in host antiviral responses. This interaction influences viral pathogenicity and host fitness. Considering its importance, sfRNA has garnered considerable attention in the field of flaviviruses since its discovery and characterization [14].

In this review, we provide an overview of sfRNA biogenesis, structures, and binding proteins in DENV. Furthermore, we compare their biological properties and functions with those of other pathogenic or non-pathogenic flaviviruses. Finally, we address some crucial questions remaining to be answered and highlight their relevance to virus interference and vaccine development.

## 2. The sfRNA of DENV Is Generated from 3′ UTR

Similar to that of other flaviviruses, the RNA genome of DENV consists of three regions based on the coding potential. There is a single large open reading frame (ORF) in the middle of the genome, encoding a large polyprotein which is further cleaved by viral and cellular proteases into three structural proteins, capsid (C), pre-membrane or membrane (prM/M) proteins, and envelope (E) proteins, and seven nonstructural proteins (NS1, NS2A, NS2B, NS3, NS4A, NS4B, and NS5). This single ORF region is flanked by UTRs which are important for effective viral translation and genome replication. The 5′ UTR is capped by a type 1 structure (m7GpppAmG). However, 3′ UTR lacks a poly (A) tail to fully mimic a cellular mRNA [18,19].

The 3′ UTR in all DENV serotypes (DENV1-4) is a highly structured region that is divided into three domains based on the conserved secondary structures. Each domain contains several cis-acting elements involved in viral RNA replication (Figure 1) [20]. Most of these secondary structures in the 3′ UTR of DENV2 have been confirmed by SHAPE biotechnology [17,21,22]. The predicted secondary structures in the 3′ UTR of DENV1, DENV3, and DENV4 still lack experimental validation. However, recent sequence covariation analysis (RNA phylogeny) of DENV 3′ UTR suggests the structures in DENV1, DENV3, and DENV4 resemble the experimentally determined structures in DENV2 [23].

### 2.1. The Roles of Domain 3 and 2 in DENV RNA Replication

Domain 3, comprising approximately the last 100 nucleotides, is the most conserved in the 3′ UTR of all DENV serotypes. It contains a short hairpin and a large stem-loop structure termed 3′ SL [24,25] (Figure 1), playing a crucial role in viral replication [26,27]. Moreover, 3′ SL is the only structure in the 3′ UTR that is absolutely needed for viral production. Keeping only the 3′ SL and replacing all other parts of the 3′ UTR with an appropriate length of poly (A) sequence in WNV RNA results in a highly attenuated but viable virus [28].

To initiate DENV RNA replication, a minus-strand RNA is first transcribed from the positive genome RNA, creating an RNase-resistant double-stranded RNA structure by remaining base-paired with the genome RNA [29]. The minus-strand RNA in this dsRNA intermediate product serves as the template for positive RNA synthesis, catalyzed by the viral RNA-dependent RNA polymerase (RdRp), NS5, in association with the viral helicase NS3 and other viral and host factors [18,30]. The 3′ SL region is indispensable for minus-strand RNA synthesis [31]. Deletion of this structure completely abolishes viral production [25,32], likely due to the disruption of long-range interaction between 5′ UTR and 3′ UTR, which is essential for NS5 recruitment [33]. Within the 3′ SL, two elements, 3′ conserved sequences (3′ CS) and the 3′ upstream AUG region (3′ UAR) (Figure 1), mediate long-range interactions by base-pairing with inverted complementary sequences in the 5′ UTR [26,27,34,35]. These interactions lead to the cyclization of the genome RNA into a conformation required for replication, bringing NS5 to the 3′ initiation site from its promoter in the 5′ UTR for minus-strand RNA synthesis [33,36]. The cyclization mechanism is also a common strategy for other flaviviruses during RNA synthesis in virus replication.

Domain 2 in all DENV serotypes contains two conserved secondary structures named dumbbells (DBs) (Figure 1), which contain the conserved sequence 2 (CS2) and repeat conserved sequence 2 (RCS2), respectively [37,38]. The DB structures include PK interactions, formed by a highly conserved motif, GCUGU, in the top loop and a downstream complementary sequence (Figure 1) [39].

The DB structures are not essential but work as enhancers for efficient virus replication, with their deletion or disruption of PK interaction resulting in decreased RNA synthesis and translation [12,34,39,40]. However, duplicated DB structures were reported to have opposite functions. The deletion of DB2 results in increased viral replication in mosquito cells, supported by the greater conservation of DB1s in different serotypes compared to DBs in the same viruses [23,41]. Since the sequence that forms PK interaction with the top loop of DB2 overlaps with 3′ CS (Figure 1), the PK interaction within DB2 competes with genome cyclization during virus replication, explaining its negative regulation of viral replication [41,42]. Conversely, DB1 is essential for genome cyclization by forming long-range interactions with a complementary sequence in the capsid region, which supplements 5′-3′ UAR and 5′-3′ CS base pairing [42,43]. These long-range interactions for genome cyclization result in mutually exclusive RNA structures during different processes [43].

In summary, the structures present in domain 3 and 2 play important roles in virus replication by regulating the genome cyclization to recruit NS5 to 3′ end of viral genome and initiate minus-strand RNA synthesis.

### 2.2. The Role of Domain 1 in DENV RNA Replication

Domain 1 displays a large heterogeneity in length, varying from 100 to 200 nt, in different serotypes (Figure 1) [44]. In DENV1-3, it contains two structurally conserved Xrn1-resistant RNAs (xrRNAs). Although this domain exhibits relatively higher genetic variability than the other two domains, the nucleotides forming the interactions within xrRNAs are conserved, with mutations in the stem region often leading to compensatory mutations for new base-pairing [17]. These xrRNAs, also known as flaviviral Nuclease-resistant RNA (fNR), resist host 5′-3′ exoribonuclease-mediated RNA degradation and are responsible for sfRNA production [14,21,23,45]. An additional base-pair interaction, a pseudoknot (PK), between the second loop of the xrRNAs and the sequence upstream of the proximal short hairpin contributes to Xrn1 resistance (Figure 1).

The xrRNA structures do not seem essential for replication, but enhance replicon RNA synthesis. The deletion of domain 1 is associated with decreased RNA synthesis and delayed virus production [34]. Despite the similar structures of the two xrRNAs in this domain, they have distinct functions in RNA replication when DENV infects mosquito or mammalian cells. Mutations in xrRNA2 benefit viral replication in C6/36 mosquito cells but are detrimental in mammalian BHK cells [17], indicating opposite selective pressures in these cell types. The different roles of xrRNAs in virus replication are attributed to their involvement in sfRNA production [16]. However, xrRNA2 is relatively less evolutionarily conserved than xrRNA1 and may have additional unknown functions [23].

In DENV4, domain 1 contains a single xrRNA phylogenetically closer to the second xrRNA in other DENV serotypes [23]. The absence of the upstream xrRNA leads to a shorter 3′ UTR (less than 400 bp) in DENV 4 compared to that of the other serotypes (around 450 bp). Although the reason for this genetic alteration is unclear, duplicated xrRNAs have been shown to enhance viral replicative fitness in both human and mosquito cells [16,17].

### 2.3. DENV sfRNA Is Generated from the 3′ UTR by the Host’s 5′-3′ Exoribonuclease, Xrn1

Generation of WNV sfRNA by cellular exoribonuclease Xrn1 was first reported by Pijlman et al. [14], and was later confirmed in other flaviviruses [21,46].

Xrn1 is an important component of the RNA turnover machinery responsible for maintaining mRNA fidelity by degrading a wide range of mRNAs and non-coding RNAs. It functions as a 5′-3′ exoribonuclease, targeting RNAs with exposed 5′ monophosphate generated by the cellular decapping complex [47]. Xrn1 also exhibits an antiviral effect on cytoplasmic RNA viruses by directly targeting and degrading viral RNAs [48,49]. In the case of DENV, Xrn1 is recruited to degrade the genome from the 5′ end to the 3′ end. However, degradation is interrupted when Xrn1 encounters xrRNAs present in the 3′ UTR, leading to the production of undegraded sfRNAs. Multiple xrRNA elements are typically present in the viral 3′ UTR, each with varying Xrn1-halting efficiencies [50], resulting in the generation of different sfRNA species. Multiple sfRNAs (sfRNA1-4) have been identified in DENV2 infection [13,16], each with a consistent 5′ end characterized by circularization and reverse transcription. In summary, sfRNA1 and sfRNA2 are produced from xrRNA1 and xrRNA2, respectively, with their 5′ ends located 5–6 nucleotides upstream of the first base-paired nucleotide in the xrRNAs; sfRNA3 and sfRNA4 are produced from DB1 and DB2, respectively, with their 5′ ends located just before the base-paired nucleotide [16], a pattern also observed in vitro [16,21].

The role of xrRNAs in Xrn1 resistance is well established. Deletions or mutations disrupting crucial interactions within the xrRNAs impairs both Xrn1 resistance and the corresponding sfRNA production [14,16,21]. While the PK interaction of xrRNA1 is functional, it is not crucial for sfRNA1 production in DENV2, as it is not detected in SHAPE analysis and has only a moderate effect on Xrn1 resistance [21]. In contrast, the small hairpin downstream of xrRNA1 seems indispensable for sfRNA1 generation [51]. Additionally, xrRNA2 in DENV2 is closely associated with xrRNA1, and mutations in xrRNA2 also reduce sfRNA1 production [16,21].

DB structures are presumably resistant to Xrn1 and are responsible for the generation of two shorter sfRNAs, although the mechanisms of Xrn1 resistance may differ from xrRNAs [21,45]. Recent research on the DB structure of Donggang virus (DONGV), an insect-specific flavivirus, suggests that DB lacks the distinctive xrRNA topology and exhibits only limited Xrn1 resistance in the presence of high concentration of magnesium in vitro [52]. The shorter sfRNAs may not be Xrn1-resistant products from DBs. Instead, they may result from trimming the 3′ end of sfRNAs produced from xrRNAs [46,52]. In addition, DB structures are crucial for coordinating with xrRNA1 in Xrn1 resistance, as the deletion of DB1 and DB2 leads to complete impairment of sfRNA1 production [12], and mutant viruses acquire duplications of xrRNA1 and xrRNA2 within the deletion region through serial passages [20]. A current study also suggests disrupting DB tertiary folding in ZIKV is associated with the decreased sfRNA1 production and cytopathic effect [40].

How DBs and xrRNA2 assist xrRNA1 in sfRNA1 production is still unknown. One possible explanation is that these individually characterized structures may form highly ordered conformations through unknown long-range RNA-RNA interactions, which may be required for sfRNA1 production. Alternatively, specific RNA-binding proteins associated with each structure may contribute to the Xrn1 resistance of xrRNA1.

## 3. The Structure of xrRNA in DENV and Other Flaviviruses

As previously mentioned, xrRNA1 and xrRNA2 in domain 1 are responsible for generating DENV sfRNA1 and sfRNA2, respectively. To comprehend how they resist Xrn1 activity, it is essential to have 3D structural information. Crystal structures of xrRNA have been resolved in Murray Valley Encephalitis Virus (MVE) [53], Zika virus (ZIKV) [54], and a few other flaviviruses [55,56]. However, the DENV xrRNA remains unavailable to date. Alignment data suggest that xrRNA structures in mosquito-borne flaviviruses are similar, with many crucial intramolecular interactions highly conserved. Therefore, we used the MVE and ZIKV xrRNA structures as models to discuss the structure-based mechanism and compared them with the predicted DENV xrRNA structure, which has been refined by SHAPE analysis [21]. We adopted the same nomenclature for each part of the structure [53,54].

### 3.1. The General xrRNA Structure

Consistent with its secondary structure, the crystal structure of xrRNA contains a three-way junction. This junction includes a “base triple” of U•A-U between single-stranded S1 and helix P3, and a “base pair” of G-C between S1 and S3, which constitutes a second pseudoknot (sPK) (Figure 2A) [50,53,54]. Nucleotides from one arm of the P1 and P3 helices form a closed ringlike structure, securely encasing the 5′ end of the xrRNA in a groove [50,53,54]. The PK interaction between the L3 loop and the S4 is absent in MVE xrRNA structure, despite the proximity of these two sequences and their tendency to pair. This indicates that PK is not an essential interaction for forming the three-way junction in MVE [53]. Consistently, mutations that impair the PK interaction result in a moderate reduction in Xrn1 resistance, which can be restored by reestablishing the interaction through co-variations. This indicates that the interaction, rather than the sequence itself, enhances the Xrn1 resistance but it is not essential [21,53]. In contrast, any mutation affecting the critical interactions within the three-way junction impairs the Xrn1 resistance of xrRNA [21,53].

Based on the structure, the “base triple” between S1 and P3 and “base pair” between S1 and S3 within the three-way junction is proposed to fold before the formation of a ringlike structure. Thus, the 5′ end of the xrRNA is encased in the structure’s center, rather than being threaded through the P1/P3 ring from the first nucleotide at the 5′ end [50]. Once the “base triple” and “base pairs” interactions occur, P3 and P1 are positioned around the xrRNA 5′ end, and the following PK interaction between L3 and S4 fully closes the P1/P3 ringlike structure. In the ZIKV xrRNA structure, this RNA is fully folded with a complete PK interaction (Figure 2A) [54].

Another long-range interaction important to fully closing the ring structure in ZIKV is formed by A37 in P3 and U51 in P1, these nucleotides being defined as the first and last nucleotide in the ring structure, respectively (Figure 2A) [54]. Disruption of this interaction has only a minor effect on xrRNA resistance in MVE. In contrast, the PK interaction is critical for both xrRNA resistance and sfRNA production in ZIKV [54]. Considering that A37/U51 and PK interactions have different effects on Xrn1 resistance in different viruses, these interactions may have distinct roles in the context of a specific xrRNA.

The mechanism for Xrn1 resistance is also proposed based on these two structures. When Xrn1 pulls RNA into the enzyme’s active center, it must unwind the P1/P3 ringlike structure. However, xrRNA may directly interact with Xrn1 to prevent its helicase activity or conformational change, keeping the 5′ end of xrRNA away from the enzyme center [53]. This model also suggests that xrRNA structure primarily prevents RNA processing events, such as degradation, from the 5′ end. Thus, the viral RNA-dependent RNA polymerase will not encounter this mechanical processing issue during the transcription of minus-strand RNA when it encounters xrRNA from the 3′ end [50].

### 3.2. xrRNA Structures in DENV

DENV xrRNAs in domain 1 belong to class 1a of xrRNAs which exists in all mosquito-borne flaviviruses, as evidenced by crystal structures in MVE and ZIKV [50,53,54,57]. Consistently, the sPK interactions formed by the “base triple” of U•A-U and the “base pair” of G-C are highly conserved across DENV serotypes (Figure 2B,C). Additionally, a conserved C would reinforce the three-way junction by forming additional hydrogen bonds with P1 to stabilize the structure (Figure 2B,C) [21]. Presumably, these structures also feature a P1/P3 ringlike structure which prevents the 5′ end of the RNA from Xrn1 degradation. The conserved A at the end of L3 and conserved U at the beginning of S4 mimic the long-range interaction seen between A37 and U51 in ZIKV, fully closing the ring structure (Figure 2B,C) [54]. While DENV xrRNA1 and xrRNA2 form distinct PK interactions, these interactions are conserved within the same xrRNA across different DENV serotypes, either as GG-CC or GAGC/U-GCUC base pairs in xrRNA1 or xrRNA2, respectively (Figure 2C). However, only the PK interaction in xrRNA2 is confirmed in SHAPE analysis [21].

Several other features characterize DENV xrRNAs. Thus, the P1 helix, comprising conserved CAGG-CCUG base pairs, is strictly required for Xrn1 resistance. Replacing them with A-U rich base pairs eliminates Xrn1 resistance. These G-C-rich base pairs in the P1 helix are also found in class 1a xrRNAs [21,50]. The P2 helix contains only one conserved base pair of C-G, and the L2 loop lacks sequence conservation. The P4 helix consists of two conserved Cs that stack with the PK interactions and may contribute to the overall conformation alteration [54]. Indeed, this helix is required for sfRNA production [51]. In summary, DENV xrRNAs form a very similar three-dimensional conformation with xrRNAs in other mosquito-borne flaviviruses.

### 3.3. xrRNA in Other Flaviviruses

Class 1a xrRNA is also identified in insect-specific flaviviruses [58], including cell-fusing agent virus (CFAV) [59], Binjari virus (BinJV) [60], Hidden Valley virus (HVV) [60] and others [60]. These flaviviruses contain one or multiple xrRNAs in the 3′ UTR and can generate different species of sfRNAs [59,60]. Although some of the conversed intramolecular interactions are experimentally confirmed using SHAPE biotechnology, crystal structures are still required to fully understand the tertiary interactions. Secondary structures indicate that xrRNAs in these viruses use a similar strategy for structure formation, but some tertiary interactions differ.

For example, the “base triple” of U•A-U within the three-way junction is replaced by C•G-C, and the conserved C between P2 and P3 is substituted with U in CFAV [59]. Additionally, novel PKs (nPKs) formed by L4 and downstream sequences are observed in these viruses [60]. These PKs do not create new Xrn1-resistant structures but only assist in the Xrn1 resistance of their upstream xrRNA. They can stall Xrn1 progression when canonical PKs are disrupted. The nPKs are only observed in ISFs and are not experimentally validated in any other flaviviruses thus far [60]. However, similar nPKs are reported in xrRNA2 of all DENV serotypes by RNA phylogeny analysis of the viral 3′ UTRs (Figure 1) [23].

Subsequently, class 2 xrRNA was identified in tick-borne flaviviruses and some arthropod vector flaviviruses [59]. These viruses, generally producing sfRNAs, contain conserved structures different from those observed in class 1a. Although both classes contain a three-way junction, class 2 features conserved structures with differing tertiary interactions within the junction. It also includes a longer P1 helix containing a bulge that serves as a halt site. Additionally, the distance between P1 and the PK interaction site in S4 is greater than in class 1a xrRNA. However, it is unknown whether these xrRNAs have three-dimensional conformation similar to those of class 1a xrRNA [59]. Given that class 2 xrRNA is only found in tick-borne flaviviruses and some arthropod vector flaviviruses phylogenetically related to tick-borne flaviviruses, it is proposed that the two classes of xrRNAs originate from different host vectors: class 1a and class 2 xrRNA are associated with mosquitoes and ticks, respectively [59].

Furthermore, xrRNA was discovered in other genera of flaviviridae based on a crystal structure in the Tamana bat virus (TABV) [55,56], termed class 1b. The structure of class 1b xrRNA highly resembles that of class 1a [55], with a more compact fold, a shorter P1 helix containing only 3 to 4 bp, and one or two non-Watson—Crick interactions at the end. It also lacks the conserved C or U between P2 and P3, which contributes to the formation of the P1/P3 ring in class 1a, as observed in the ZIKV structure [54]. However, the more compact structure of class 1b does not tolerate or require this nucleotide. Additionally, the length and pattern of PK interaction are more conserved in the class 1b structure. No virus species are known to contain both class 1a and 1b sequences, indicating that these sequences likely originated from common ancestors rather than being transferred from other viral species [56].

In summary, xrRNAs are found throughout the Flaviviridae family, but many aspects of their Xrn1 resistance, role in sfRNA production, and impact on viral pathogenicity remain unclear. However, the development and preservation of these structures during the long term of evolution suggest that they represent common strategies for interacting with hosts and overcoming antiviral effects in all Flaviviridae members.

## 4. Proteins Associated with sfRNAs

Non-coding RNAs which are not translated into functional proteins exert their regulatory functions through interactions with various host cellular proteins. These proteins directly bind to the non-coding RNA and regulate its biogenesis, structure, expression, subcellular localization, and interactome. As DENV and other flavivirus sfRNAs are virus-derived non-coding RNAs, their primary mechanism of action in host antiviral responses and viral pathogenicity is through interactions with both viral and host-binding proteins. Indeed, numerous binding proteins have been reported in infections with DENV or other flaviviruses (Table 1). Considering that the 3′ UTR shares a similar structure with sfRNA, we also summarize the binding proteins associated with the 3′ UTR (Table 1).

Using in vitro transcribed RNA of 3′ UTR as the bait, RNA pulldown assays combined with mass spectrometry analysis are the general approaches to identify the associated proteins of sfRNAs [61,62,63,64,65,66,67,68,69]. Similar experiments followed by Western blot could further identify specific determinants by a series of deletions of RNA secondary structures [64,65,66,67,68]. Recent advancement of proximity biotinylation in pulldown assays enables identifying transient RNA and protein interactions of direct or indirect sfRNA binding proteins [70]. In addition to RNA pulldown assays, the three-hybrid method in mammalian cells (RNA-KISS) and yeast cells (Y3H) is also a valuable approach to complement the sfRNA interactome. Nonetheless, the functions of these novel binding proteins in flavivirus infection and pathogenesis remain unknown [71,72]. The DEAD-Box Helicase 6, DDX6, characterized as DB structure binding protein [65], has been identified in many RNA pulldown assays for DENV or ZIKV sfRNA in both mammalian and mosquito cells [61,66,68], as well as three-hybrid screening [71,72]. However, its roles in different flaviviruses are controversial [65,66,68]. DDX6 is an essential component of processing bodies (PBs) involved in mRNA storage and turnover [73]. It is recruited to the DENV replication center [65]. Given that DENV and other flaviviruses may interfere with PB assembly [74], the connection between DDX6, PBs, and flavivirus replication needs further investigation.

**Table 1 viruses-15-02306-t001:** Proteins associated with sfRNAs or 3′ UTR.

Protein Symbol	Methods	Bait/Target	Extract/Prey	Binding Structure	Functions
TRIM25 MAVS [62]	RNA (in vitro transcribed) pull-down and MS	DENV2 3′ UTR	HuH-7 cells (Human)	Unknown	DENV sfRNAs bind to TRIM25 to interfere with RIG-I-mediated antiviral signaling [62].
YBX1 HNRNPA1 HNRNPA2B1 SYNCRIP HNRNPH1 [63]	RNA (in vitro transcribed) pull-down and MS	DENV2 3′ UTR	BHK cells (Hamster)	YBX1: 3′ SL	YBX1 restricts DENV replication and translation [63].
DDX6 CAPRIN1 G3BP1 G3BP2 USP10 [64,65]	RNA (in vitro transcribed) pull-down and MS	DENV2 3′ UTR	HeLa cells (Human)	DDX6: DB1 and DB2 CAPRIN1, G3BP1, and G3BP2: 3′ UTR (xrRNA1 required)	DENV2 sfRNA sponges G3BP1, G3BP2, and CAPRIN1 to downregulate the translation of ISGs * [64]; DDX6 facilitates DENV replication [65].
DDX6 EDC3 PHAX SF3B1 APOBEC3C [66]	RNA (in vitro transcribed) pull-down and MS	DENV2 and ZIKV 3′ UTR	HeLa cells (Human)	Unknown	ZIKV sfRNA sequesters RNA splicing factors to dysregulate host RNA processing; DDX6 restricts ZIKV replication [66].
FMR1 FXR1 FXR2 [67]	RNA (in vitro transcribed) pull-down and MS	ZIKV 3′ UTR	JEG3 cells (Human)	xrRNA1	ZIKV sfRNA suppresses the antiviral effects of FMRP; FMRP restricts ZIKV infection by inhibiting viral translation [67].
ME31B (DDX6) ATX2 LSM12 AAEL018126 [68]	RNA (in vitro transcribed) pull-down and MS	ZIKV and DENV 3′ UTR	Aag2 cells (mosquito)	ME31B (DDX6): DB2	ZIKV sequesters PB components, including DDX6 (ME31B), ATX2, LSM12, and AAEL018126; ME31B restricts ZIKV replication in Ae. aegypti cells [68].
AGO2 DICER [75]	RNA immunoprecipitation	Kunjin virus 3′ UTR	293T cells (Human)	Unknown	The sfRNAs sequester Dicer and Ago2 to suppress RNA interference [75].
NS5 [69]	RNA (in vitro transcribed) pull-down and MS	ZIKV 3′ UTR	A549 cells (Human)	Unknown	The sfRNAs interact with NS5 to inhibit the IFN signaling [69].
AeMaleless AeSex-lethal AeGTPase AeStaufen AAEL001518 AAEL004834 AeRan AeDIP1 AeRpS24 AAEL014376 AeRNase AeExoRNase AeAtu AePur (DDX6) [61]	RNA (in vitro transcribed) pull-down and MS	DENV2 3′ UTR	Aag2 cells (mosquito)	Unknown	AeStaufen reduces the secretion of sfRNA in saliva [61].
Sec61A1 Loquacious [70]	RNA-protein interaction (RAPID) assay	Domain 2 and 3 of DENV2 3′ UTR	C6/36 cells (mosquito)	Unknown	Sec61A1 and Loquacious promote DENV replication [70].
Refer to [72]	Yeast three-hybrid (Y3H) screening assay	DENV2 and ZIKV 3′ UTR	human ORF library	Unknown	Unknown
Refer to [71]	Kinase Substrate Sensor (KISS)	DENV 3′ UTR	human ORF library	Unknown	Unknown

* 3′ UTR, 3′ untranslated region; DENV, Dengue virus; MS, mass spectrometry; IFN, Interferon; ISGs, interferon-stimulated genes; sfRNAs, Subgenomic flavivirus RNAs; RIG-I, Retinoic acid-inducible gene I; ZIKV, Zika virus.

## 5. Function of the sfRNAs in DENV

The generation of sfRNA serves as a vital viral strategy to evade host antiviral responses in both arthropod vectors and vertebrate hosts. This is substantiated by two lines of compelling evidence. First, sfRNA acts as an evolved degradation product that counters the host antiviral exoribonuclease, Xrn1 [48,49]. Second, it directly suppresses antiviral responses by sequestering proteins crucial for the activation of interferon (IFN) responses [62], the translational regulation of interferon-stimulated genes (ISGs) [64], components of the RNA interference (RNAi) pathway [75], and other antiviral proteins [63,67,68]. The fundamental role of sfRNAs is to shape the virus’s replicative fitness, epidemiological fitness, and transmission [16,17,62,76,77].

### 5.1. sfRNAs Suppress Antiviral Responses

sfRNAs inhibit type I IFN response in infections caused by WNV and JEV, although the underlying mechanisms are still unclear [78,79]. One proposed mechanism underlying the inhibitory effect of sfRNAs on IFN response involves the interaction of DENV sfRNAs with specific RNA binding proteins, namely G3BP1, G3BP2, and CAPRIN1, dependent on the xrRNA1 structure. DENV sfRNAs bind to these three RNA binding proteins and antagonize their functions in ISG translation, resulting in a reduced IFN response during DENV infection [64].

An alternative mechanism was demonstrated in the study of epidemic DENV isolates form Puerto Rico [62,80]. In cases of high epidemic DENV strains, an accumulation of sfRNAs was associated with low expression of IFN-β. This indicates that sfRNAs from high epidemic DENV strains regulate virus epidemiological fitness through inhibition of type I IFN response. In vitro transcribed sfRNAs from high epidemic DENV strains reduce Poly IC-triggered IFN expression [62]. In contrast, sfRNAs from low epidemic DENV exhibit a lower effect on IFN expression [62]. sfRNAs from both high and low epidemic DENV strains have been shown to directly interact with TRIM25 preventing RIG-I-mediated type I IFN induction. However, high epidemic DENV sfRNAs exhibited a higher affinity for complex formation with TRIM25 than low epidemic counterparts. Consequently, high epidemic DENV sfRNAs effectively reduce IFN responses, ensuring high virus production and transmission [62].

Conversely, a recent study involving ZIKV sfRNAs revealed a distinct mechanism. ZIKV sfRNAs were found not to inhibit the expression of IFN but to interfere with downstream IFN signaling [69]. Genome-wide gene expression profiling and pathway analysis revealed that type I IFN signaling but not IFN expression are enhanced in sfRNA-deficient ZIKV mutants. Depletion of sfRNAs in ZIKV mutants led to enhanced type I IFN signaling due to the inability of the mutants to inhibit the phosphorylation of STAT1, a crucial downstream transcription factor in IFN pathway. This mechanism was shown to be dependent on the binding of NS5 to the sfRNAs [69]. This study also pointed out that the effect of sfRNAs on IFN signaling may be very different within the mosquito-borne flaviviruses based on the distinct context of sequences or structures in the sfRNAs.

The sfRNAs have also been reported to target RNAi response, a prominent antiviral mechanism in insects and plants [68,75,81,82]. Although the exact mechanism through which sfRNAs influence RNAi activity is still a topic of ongoing research, it is possible that they sequester either important components from RNAi machinery [75] or P bodies [68] to repress RNAi activity.

### 5.2. Duplication of xrRNAs Contributes to DENV Replicative Fitness

The structural stability of the xrRNAs in the DENV 3′ UTR is crucial for sfRNA production and efficient viral replication. Their structures are relatively stable, and mutations are often compensated by new base-paired co-variations, maintaining their structural integrity. Nevertheless, xrRNA2 in DENV2 strains isolated from mosquitoes frequently acquires mutations which may disrupt its structure [17,23]. These mutations are particularly advantageous for DENV2 replication in mosquito cells, resulting in increased replication [17]. However, xrRNA2 destabilizing mutations are less favorable for replication in mammalian cells, underscoring the critical role of xrRNA1 in DENV2 replication in the presence of these mutations [17]. Introducing xrRNA2-destabilizing mutations drastically reduces viral replication in mammalian cells when the virus only contains xrRNA2 [17]. Based on these observations, duplicated xrRNAs might be critical for providing higher tolerance against detrimental mutations during host switching by preserving at least one intact xrRNA [42].

Further study demonstrated that DENV produces different patterns of sfRNAs in mosquito and human cells due to adaptive mutations in the xrRNA2 region [16]. The human-adapted virus mainly generates sfRNA1 with a minor amount of sfRNA2. Conversely, the mosquito-adapted virus containing specific xrRNA2 mutations mainly generates species of sfRNAs (sfRNA3 and sfRNA4) [16]. The mosquito-adapted virus induces a higher level of expression of IFN and ISGs in human cells. This effect is attributed to the accumulation of shorter sfRNA3 and sfRNA4, which are unable to counteract the IFN response or directly trigger RIG activation [16]. Thus, the generation of shorter sfRNA species is linked to reduced viral replicative fitness in mammalian cells.

Based on the above observations, duplicated xrRNAs which maintain long sfRNAs seem critical for the virus to cycle between the hosts, and the lack of duplicated xrRNAs in ISFs results in host restriction. However, this theory faces challenges from recent findings [60]. Both insect-specific and mosquito-borne flaviviruses contain duplicated xrRNAs in their 3′ UTR. This duplication is just a backup strategy to ensure sfRNA production when the upstream xrRNA loses Xrn1 resistance due to misfolding or critical mutations [60]. The redundancy of duplicated xrRNAs is also supported by observations that single sfRNA-deficient Zika and West Nile viruses replicate similarly to the wild-type virus in both mosquito and mammalian cells. However, the depletion of both sfRNAs typically abolishes virus replication [60,69]. Although the function of duplicated xrRNAs in virus replicative fitness has been reported primarily for DENV2, further research is needed to explore the correlation between duplicated xrRNAs and virus fitness.

### 5.3. High Level of sfRNAs Facilitate Virus Transmission

The accumulation of high levels of sfRNAs in mosquito salivary glands infected with epidemic isolates of DENV2 indicates a link between sfRNAs and viral transmission. Substitutions within the 3′ UTR between high and low epidemic virus isolates are responsible for the elevated levels of sfRNAs and viral titers in salivary glands [77]. In addition, high epidemic virus isolates are associated with a low level of Toll immune signaling pathway. Given that sfRNAs interact with TRIM25 to repress RIG-I-mediated antiviral response [62], the accumulated sfRNAs are thought to enhance the infection rate as an immune suppressor [77]. Indeed, the sfRNAs are loaded into extracellular vesicles and released into saliva, potentially compromising local innate immune responses when delivered into the mammalian cells during mosquito biting [76]. Thus, a high level of sfRNAs facilitates virus transmission, ultimately benefiting DENV epidemiology [76,77]. Consistently, sfRNAs have also been found to be essential for overcoming the midgut barrier during virus transmission in WNV infection [83].

However, recent findings in ZIKV infections suggest that sfRNA-deficient ZIKV efficiently infects mosquito salivary glands but fails to release into saliva [84], indicating a new mechanism underlying reduced virus transmission in the absence of sfRNA. In this context, the Toll immune signaling pathway remains unaffected [68,84], but sfRNA-deficient virus induces caspase-7 expression and apoptosis in infected mosquito tissues [84], while also elevating the siRNA response [68]. These observations suggest that ZIKV sfRNAs inhibit apoptosis and the RNAi response in mosquitoes, facilitating virus infection and transmission [68,84]. Consequently, sfRNAs employ different strategies to enhance virus transmission in various mosquito-borne flaviviruses.

sfRNAs may exhibit distinct functions in mosquito cells compared to mammalian cells, as seen in DENV infections where they induce apoptosis through the Bcl-2-mediated PI3k/Akt pathway. Consequently, further research is needed to clarify the specific roles of sfRNAs in mosquito-borne flaviviruses [12].

## 6. Perspectives

DENV sfRNA is generated from the 3′ UTR and plays inhibitory roles in the antiviral response; thus, it is an attractive target for vaccine design and other antiviral strategies. sfRNA structure modification without disrupting the viral intact coding potentials and antigenic specificity may be a feasible approach to generate live-attenuated vaccine candidates for DENV and other flaviviruses. For example, the candidate vaccine, rDEN4Δ30, is attenuated by a 30 nt deletion (Δ30) in DB1 of DENV4 [85]. Candidate vaccines of the other DENV serotypes have also been developed by similar strategy, and a single dose of tetravalent admixtures could induce protection against all DENV serotypes [86]. Moreover, replacements of prM and E gene region in DEN4Δ30 backbone with analogous sequences from WNV or ZIKV produce live attenuated chimeric vaccines, WN/DEN4Δ30 [87] or rZIKV/D4Δ30-713 ZIKV [88], respectively. Recently, the analogous 10 nt deletion in an infectious cDNA clone of ZIKV also generates attenuated candidate strain with high immunogenicity and safety [89]. The molecular mechanism of this effective attenuation strategy has been linked to the decreased sfRNA production and increased IFN susceptibility [90]. Therefore, impaired sfRNA production or function should be considered for other antiviral strategies. This concept suggests that a detailed understanding of the tertiary structure of xrRNAs is essential. Targeting the crucial intramolecular interactions within these structures through mutations or deletions could potentially disrupt its resistance to Xrn1.

In summary, although the biogenesis and functions of DENV sfRNA have been well characterized in virus-infected cells, several key questions remain. First, the precise subcellular localization of DENV sfRNA, from its generation to exerting its function, needs further investigation. sfRNAs share the same sequence with the viral 3′ UTR, making it hard to differentiate them using traditional RNA-fluorescence in situ hybridization. However, ectopic expression of sfRNAs or transfection of in vitro transcribed sfRNA without DENV infection may not fully mimic the natural localization of sfRNAs. Thus, a more specific method of sfRNA detection or visualization is required to know whether its generation is coupled with virus replication in the replication center or happens in other cell compartments after viral RNA synthesis and delivery. Second, a comprehensive understanding of the interactome of DENV sfRNA requires the development of a virus-derived sfRNA purification-based pull-down approach to avoid potential artifacts. Third, exploring potential RNA modifications on DENV sfRNAs, especially within crucial structures, in the context of natural DENV infections is of interest to determine whether sfRNAs undergo post-transcriptional regulation by virus and host RNA modification enzymes.

## Figures and Tables

**Figure 2 viruses-15-02306-f002:**
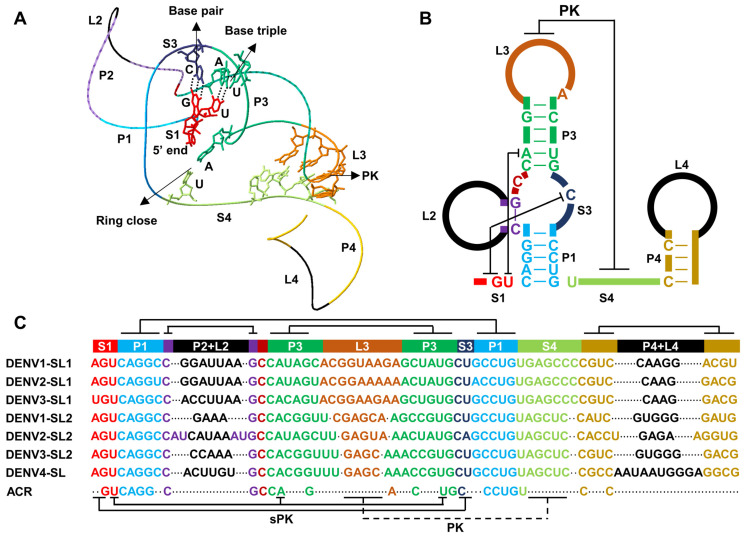
Dengue virus (DENV) xrRNAs belong to Class 1a xrRNA. (**A**) The 3D structure of Zika virus (ZIKV) xrRNA (PDB accession number 5TPY) is visualized using the Swiss PDB Viewer with colored stems and loops. The important base pairs within the three-way junction and ringlike structure and the unformed pseudoknot (PK) interaction are shown. (**B**) Secondary structure of the xrRNAs in DENV, with stems and loops in the same colors, and the conserved nucleotides. PK and second (s)PK interactions corresponding to that in MVE and ZIKV xrRNAs are indicated by black lines. (**C**) Alignments of DENV xrRNAs, with stems and loops in the same colors. The base-paired sequences corresponding to that in MVE and ZIKV xrRNAs are indicated by black lines. ACR, absolutely conserved region.

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
