# Peer review of "Subgenomic Flaviviral RNAs of Dengue Viruses"

_viruses, 2023, doi:10.3390/v15122306_

Round 1

Reviewer 1 Report

Comments and Suggestions for Authors

                This manuscript represents a well written and rather comprehensive overview of sfRNAs made by dengue viruses (and other insect-borne flaviviruses).  The topic has been previously covered in more than a dozen previously published reviews, but this is well-written and may be of interest to some readers.  I do have a few minor suggestions to polish the manuscript:

Minor Points:

1.        Line 46:  I would delete the sentence ‘No proteins were identified as translated from this region until now.’ It is redundant with the previous sentence and the ‘until now’ part may imply to some readers that there are proteins made form sfRNA that have been recently identified.

2.       Line 159-160:  Xrni is not ‘targeting RNAs with exposed 5′ caps generated by the cellular decapping complex’ – it targets an exposed 5’ monophosphate generated by decapping.

3.       Table 1:  While this table describes sfRNA binding proteins that have been most extensively characterized, the RNA pulldowns in the cited references identified numerous other proteins that were not the focus of follow up studies.  The authors may wish to comment on the pulldown results as well to make this section a bit more comprehensive for the reader.

4.       Line 430:  The f in sfRNA appears to be underlined.

5.       To make the review as up to date as possible, the authors may wish to mention the findings of the recent paper by Graham et al (mBio. 2023 Aug 31;14(4):e0110823. doi: 10.1128/mbio.01108-23.)

Author Response

Point-by-point responses to the reviewers’ comments (viruses-2706152)

Reviewer 1

Comments and Suggestions for Authors

                This manuscript represents a well written and rather comprehensive overview of sfRNAs made by dengue viruses (and other insect-borne flaviviruses).  The topic has been previously covered in more than a dozen previously published reviews, but this is well-written and may be of interest to some readers.  I do have a few minor suggestions to polish the manuscript:

Minor Points:

  1. Line 46: I would delete the sentence ‘No proteins were identified as translated from this region until now.’ It is redundant with the previous sentence and the ‘until now’ part may imply to some readers that there are proteins made form sfRNA that have been recently identified.

Response:

Line 45: The sentence “‘No proteins were identified as translated from this region until now” has been deleted according to the reviewer’s suggestion.

  1. Line 159-160: Xrni is not ‘targeting RNAs with exposed 5′ caps generated by the cellular decapping complex’ – it targets an exposed 5’ monophosphate generated by decapping.

Response:

Line 157: “5’ caps” has been corrected and replaced by “5’ monophosphate”.

  1. Table 1: While this table describes sfRNA binding proteins that have been most extensively characterized, the RNA pulldowns in the cited references identified numerous other proteins that were not the focus of follow up studies.  The authors may wish to comment on the pulldown results as well to make this section a bit more comprehensive for the reader.

Response:

Line 336-352: As suggested by the reviewer, we discussed the general approaches to identify sfRNA binding proteins and further emphasized on DDX6 protein.

  1. Line 430: The f in sfRNA appears to be underlined.

Response:

Line 445: The underline has been deleted.

  1. To make the review as up to date as possible, the authors may wish to mention the findings of the recent paper by Graham et al (mBio. 2023 Aug 31;14(4):e0110823. doi: 10.1128/mbio.01108-23.)

Response:

Line 112: the paper is cited.

Line 189-190: a statement of “A current study also suggests disrupting DB tertiary folding in ZIKV is associated with the decreased sfRNA1 production and cytopathic effect” has been added in the revised manuscript.

Reviewer 2 Report

Comments and Suggestions for Authors

General comments:

-          This is a valuable review article. It aims at reviewing and discussing DENV sfRNA structures and functions.

 -          However, this manuscript might need to be restructured. The information regarding the RNA structures are dispersed throughout the manuscript. For instance, the three-way junction RNA (xrRNA) is presented in section 2.1, its function is described in section 2.3 and it structure is shown in 3.  For the expert reader, it will be possible to connect these topics. For a newcomer to the field, it may be difficult. It will be easier for the reader to structure this review article following the functional and structural domains in the DENV sfRNA (or DENV 3’UTR):  Domain I, II and III (2D and 3D RNA structures, evolution, and functions). An alternative option can be describing functional and structural domains, briefly discussing the role of structures and functions in Domain III and II and subsequently focusing on the domain I, RNA structure, sfRNA production, function, and evolution.

 -          An extension of the “perspective” section on the role of sfRNA and 3’UTR deletions on dengue and flavivirus vaccine development will be valuable to highlight the relevance of understanding this segment of DENV or any flavivirus genome. Some of the cited articles refer to this topic. Other articles should be cited to refer to ZIKV, WNV, DENV and flaviviruses' live attenuated vaccines (with 3'UTR deletions). 

Specific comments:

-          The DENV sfRNA secondary structures in figure 1 do not reflect the current understanding of these RNA structures:

 o   Please avoid using the “SL” term to refer to the structures in Domain I. They are three-way-junctions, not Stem-Loop (SL) structures. The use of the “SL” term only reflected the lack of understanding of these RNA structures in the past. The biochemical term Xrn1-resistant RNA (xrRNA) has been suggested in reference 22 and 52.  A third term, the “flaviviral Nuclease-resistant RNA (fNR)” is suggested in reference 23. The fNR term invoked the functional and evolutionary role of these RNA structures. Any of these two terms will better reflect the current state-of-the-art in the sfRNA field.

o   The three-way junction structures in Domain I are not depicted as such in Fig 1. A good template for depicting the structures in DENV sfRNA is provided in reference 22 and 23.

o   The fifth pseudoknot reported in reference 23 is also lacking.

o   “srRNA” is written on the DENV2 sfRNA in Figure 1. It should be “sfRNA”.

o   The CS2 and RCS2 sequences in domain II are mentioned in the main text (line 87 and 88), they must be shown or highlighted in Fig 1.

-          Comment on Lines 94-96. The authors cite an article on West Nile Virus attenuated strains. They should clarify that this knowledge was obtained from work on a distinct flavivirus, not dengue viruses.

-          Comment on Line 97-100: It should also be noted that the sequence covariation analysis (also known as RNA phylogeny) in reference 23 suggested that the RNA structures in DENV1, 3 and 4 sfRNA are evolutionarily linked homologues to those experimentally determined RNA structures in DENV2 and they can be regarded as valid as the DENV2 RNA structures. In the same way as the RNA phylogeny of few ribosomal RNA sequences (performed in the 1970’s) allowed us to know the global fold of RNA structures in all ribosomes.

-          The role of DB structures in Domain II of DENV3’UTR on DENV translation were not discussed in section 2.2.

-          Figure 2A shows the MVEV xrRNA crystal structure. This crystal structure didn’t show the pseudoknot. The ZIKV xrRNA will be a better crystal structure to show (PDB: 5TPY).

-          Section 4 should be further discussed and the identified sfRNA binding proteins should be shown and discussed separately for mosquitoes and human proteins.

-          Section 5 is pristine. It nicely summarizes the recent advancement and current understanding of the field.

Comments on the Quality of English Language

Some sentences make me pause the reading. They are not wrong. It often means that the verb is very distant from the noun.

Author Response

Point-by-point responses to the reviewers’ comments (viruses-2706152)

Reviewer 2

Comments and Suggestions for Authors

General comments:

-          This is a valuable review article. It aims at reviewing and discussing DENV sfRNA structures and functions.

 -          However, this manuscript might need to be restructured. The information regarding the RNA structures are dispersed throughout the manuscript. For instance, the three-way junction RNA (xrRNA) is presented in section 2.1, its function is described in section 2.3 and it structure is shown in 3.  For the expert reader, it will be possible to connect these topics. For a newcomer to the field, it may be difficult. It will be easier for the reader to structure this review article following the functional and structural domains in the DENV sfRNA (or DENV 3’UTR):  Domain I, II and III (2D and 3D RNA structures, evolution, and functions). An alternative option can be describing functional and structural domains, briefly discussing the role of structures and functions in Domain III and II and subsequently focusing on the domain I, RNA structure, sfRNA production, function, and evolution.

Response:

Section 2 has been revised as the reviewer suggested. In the current manuscript, section 2.1 discusses the structures of domain 3 and 2, and their functions in virus RNA replication and translation. Section 2.2 is focused on the domain 1 structure, function and evolution.

 -          An extension of the “perspective” section on the role of sfRNA and 3’UTR deletions on dengue and flavivirus vaccine development will be valuable to highlight the relevance of understanding this segment of DENV or any flavivirus genome. Some of the cited articles refer to this topic. Other articles should be cited to refer to ZIKV, WNV, DENV and flaviviruses' live attenuated vaccines (with 3'UTR deletions).

Response:

Line 467-477: As suggested by the reviewer, articles referred to ZIKV, WNV, DENV and flaviviruses' live attenuated vaccines have been discussed and cited.

Specific comments:

-          The DENV sfRNA secondary structures in figure 1 do not reflect the current understanding of these RNA structures:

 o   Please avoid using the “SL” term to refer to the structures in Domain I. They are three-way-junctions, not Stem-Loop (SL) structures. The use of the “SL” term only reflected the lack of understanding of these RNA structures in the past. The biochemical term Xrn1-resistant RNA (xrRNA) has been suggested in reference 22 and 52.  A third term, the “flaviviral Nuclease-resistant RNA (fNR)” is suggested in reference 23. The fNR term invoked the functional and evolutionary role of these RNA structures. Any of these two terms will better reflect the current state-of-the-art in the sfRNA field.

Response:

All the “SL”s have been replaced by “xrRNA” in the revised manuscript.

Line 132-133: The alternative name, flaviviral Nuclease-resistant RNA (fNR), is also mentioned with reference 23 cited.

o   The three-way junction structures in Domain I are not depicted as such in Fig 1. A good template for depicting the structures in DENV sfRNA is provided in reference 22 and 23.

Response:

Figure 1: the structures have been corrected in the new figure 1.

o   The fifth pseudoknot reported in reference 23 is also lacking.

Response:

Figure 1: the fifth pseudoknot has been indicated as novel PKs in the new figure.

Line 295-297: a statement of “similar nPKs are reported in xrRNA2 of all DENV serotypes by RNA phylogeny analysis of the viral 3’ UTRs” has been added in the revised manuscript.

o   “srRNA” is written on the DENV2 sfRNA in Figure 1. It should be “sfRNA”.

Response:

Figure 1: “srRNA”s have been corrected in the new figure.

o   The CS2 and RCS2 sequences in domain II are mentioned in the main text (line 87 and 88), they must be shown or highlighted in Fig 1.

Response:

Figure 1: The CS2 and RCS2 have been indicated in the new figure.

-          Comment on Lines 94-96. The authors cite an article on West Nile Virus attenuated strains. They should clarify that this knowledge was obtained from work on a distinct flavivirus, not dengue viruses.

Response:

Line 88: WNV has been mentioned in the sentence in the revised manuscript.

-          Comment on Line 97-100: It should also be noted that the sequence covariation analysis (also known as RNA phylogeny) in reference 23 suggested that the RNA structures in DENV1, 3 and 4 sfRNA are evolutionarily linked homologues to those experimentally determined RNA structures in DENV2 and they can be regarded as valid as the DENV2 RNA structures. In the same way as the RNA phylogeny of few ribosomal RNA sequences (performed in the 1970’s) allowed us to know the global fold of RNA structures in all ribosomes.

Response:

Line 79-81: A sentence of “However, recent sequence covariation analysis (RNA phylogeny) of DENV 3’ UTR suggests the structures in DENV1, DENV3, and DENV4 resemble the experimentally determined structures in DENV2” has been added in the revised manuscript. Reference 23 is cited.

-          The role of DB structures in Domain II of DENV3’UTR on DENV translation were not discussed in section 2.2.

Response:

Line 112: Deletion of both DB structures reduced DENV2 translation has been reported by reference 34. We have revised the manuscript accordingly.

-          Figure 2A shows the MVEV xrRNA crystal structure. This crystal structure didn’t show the pseudoknot. The ZIKV xrRNA will be a better crystal structure to show (PDB: 5TPY).

Response: Considering that MVEV xrRNA is the first resolved crystal structure and contains the most important intramolecular interactions within the three-way junction, we would like to present and discuss this structure first. And the fully closed ring structure and pseudoknot will be discussed by using ZIKV xrRNA structure.

-          Section 4 should be further discussed and the identified sfRNA binding proteins should be shown and discussed separately for mosquitoes and human proteins.

Response:

Table 1: The extract from human or mosquito cells were shown in Table 1.

Line 336-352: A new paragraph was added to discuss the general approaches to identify sfRNA binding proteins and further emphasized on DDX6 protein.

-          Section 5 is pristine. It nicely summarizes the recent advancement and current understanding of the field.

Response:

Thanks.

Comments on the Quality of English Language

Some sentences make me pause the reading. They are not wrong. It often means that the verb is very distant from the noun.

Response:

Some long sentences have been rewritten as suggested.

Line 99-100:

Original sentence, “Within the 3’ SL, two elements, 3’ conserved sequences (3’ CS) immediately upstream of the short hairpin, and the 3’ upstream AUG region (3’ UAR) overlapping with both the short hairpin and large stem (Figure 1), mediate long-range interactions by base-pairing with inverted complementary sequences in the 5’ UTR”.

Rewritten sentence, “Within the 3’ SL, two elements, 3’ conserved sequences (3’ CS), and the 3’ upstream AUG region (3’ UAR) (Figure 1), mediate long-range interactions by base-pairing with inverted complementary sequences in the 5’ UTR”.

Line 153-154:

Original sentence, “Generation of sfRNA in the cytoplasm of DENV from viral genome RNA by cellular exoribonuclease Xrn1 was first reported by Pijlman et al., and was later confirmed in other flaviviruses”.

Rewritten sentence, “Generation of WNV sfRNA by cellular exoribonuclease Xrn1 was first reported by Pijlman et al., and was later confirmed in other flaviviruses”.

Line 171-173:

Original sentence, “The role of xrRNAs in Xrn1 resistance is well established, with deletions or muta-tions disrupting crucial interactions within the xrRNAs and impairing both Xrn1 re-sistance and the corresponding sfRNA production”.

Rewritten sentences, “The role of xrRNAs in Xrn1 resistance is well established. Deletions or mutations disrupting crucial interactions within the xrRNAs impairs both Xrn1 resistance and the corresponding sfRNA production”.

Round 2

Reviewer 2 Report

Comments and Suggestions for Authors

One comment:
Authors must change the MVEV structure for the ZIKV crystal structure in Figure 2. Not doing so implies ignoring the current understanding of all the contacts in the three way junction RNA structure, and it is a complete disregard to the major effort from the Kieft group to obtain a crystal structure with this feature and the associated additional interactions.

Author Response

Comments and Suggestions for Authors

One comment:
Authors must change the MVEV structure for the ZIKV crystal structure in Figure 2. Not doing so implies ignoring the current understanding of all the contacts in the three way junction RNA structure, and it is a complete disregard to the major effort from the Kieft group to obtain a crystal structure with this feature and the associated additional interactions.

Response:

The figure has been replaced by ZIKV xrRNA structure  (PDB accession number 5TPY), and the manuscript was revised accordingly.